# Tensile Bond Strength between Different Denture Base Materials and Soft Denture Liners

**DOI:** 10.3390/ma16134615

**Published:** 2023-06-26

**Authors:** Josip Vuksic, Ana Pilipovic, Tina Poklepovic Pericic, Josip Kranjcic

**Affiliations:** 1Department of Removable Prosthodontics, School of Dental Medicine, University of Zagreb, Gunduliceva 5, 10000 Zagreb, Croatia; jvuksic@sfzg.hr; 2Department of Prosthodontics, University Hospital Dubrava, Av. Gojka Šuška 6, 10000 Zagreb, Croatia; 3Department of Technology, Faculty of Mechanical Engineering and Naval Architecture, University of Zagreb, Ivana Lučića 5, 10000 Zagreb, Croatia; ana.pilipovic@fsb.hr; 4Department of Prosthodontics, School of Medicine, University of Split, Šoltanska 2, 21000 Split, Croatia; tinapoklepovic@gmail.com; 5Department of Fixed Prosthodontics, School of Dental Medicine, University of Zagreb, Gunduliceva 5, 10000 Zagreb, Croatia

**Keywords:** denture liners, CAD-CAM, denture base

## Abstract

(1) Background: Various materials are available for CAD-CAM denture base fabrication, for both additive and subtractive manufacturing. However, little has been reported on bond strength to soft denture liners. Therefore, the aim of this study was to investigate tensile bond strength, comparing between different denture base materials and soft denture liners. (2) Methods: Seven different materials were used for denture base fabrication: one heat-polymerized polymethyl methacrylate, three materials for subtractive manufacturing, two materials for additive manufacturing and one polyamide. Two materials were used for soft denture lining: one silicone-based and one acrylate-based. The study was conducted according to the specification ISO No. 10139-2:2016, and the type of failure was determined. The Kruskal–Wallis test with Dunn’s post hoc test was used to analyse the values of tensile bond strength, and Fisher’s exact test was used to analyse the type of failure. *p* Values < 0.05 were considered statistically significant. (3) Results: The tensile bond strength values were not statistically significantly different combining all the materials used for denture base fabrication with the acrylate-based soft denture liner (*p* > 0.05), and the average values ranged between 0.19 and 0.25 Mpa. The tensile bond strength values of the different denture base materials and silicone-based denture liner were statistically significantly different (*p* < 0.05), and the average values ranged between 1.49 and 3.07 Mpa. The type of failure was predominantly adhesive between polyamide and both additive-manufactured denture base materials in combination with the acrylate-based soft liner (*p* < 0.05). (4) Conclusions: The use of digital technologies in denture base fabrication can have an influence on different tensile bond strength values for soft denture liners, with different types of failure when compared with heat-cured PMMA. Similar tensile bond strength values were found between the acrylate-based soft denture liner and denture base materials. Significant differences in tensile bond strength values were found between the silicone-based soft denture liner and denture base materials, where the additive-manufactured and polyamide denture base materials showed lower values than heat-cured PMMA and subtractive-manufactured denture base materials.

## 1. Introduction

Among the various materials used for denture base fabrication, polymethyl methacrylate (PMMA) became the gold standard soon after its introduction into clinical use [1,2]. It has many advantageous properties, including a low cost, ease of handling, a light weight, low water solubility and water sorption, stability in oral environments and high aesthetic results, but also has some shortcomings, including a residual monomer, brittleness, poor mechanical properties, high polymerization shrinkage and a lack of radiopacity [1,3,4,5]. For this reason, there is an ongoing search for a better material. One direction is to chemically modify PMMA using monomers, oligomers, copolymers and cross-linking agents [6,7,8,9]. Investigations were also performed with the incorporation of filler particles and fibres into the PMMA, with recent trends towards nanoparticle incorporation (zirconium dioxide nanoparticles, silicone dioxide nanoparticles, diamond nanoparticles) [5,9]. Different processing techniques were also proposed (injection moulding, microwave, heat polymerization under high pressure, autoclave) [9]. The other direction is the use of materials with a completely different chemical formula from PMMA, such as polyamide, polycarbonate and polyester [10,11]. When compared with PMMA, these materials have a lower elastic modulus, lower surface roughness, lower allergenic risk and higher resistance to acids, but also have some other disadvantages, including complicated manipulation and different processing and polishing methods, the fact that special and more expensive equipment is required, higher water sorption, a higher risk of fracture and lower colour stability [10,11]. Recently, polyetheretherketone (PEEK) and a high-performance polymer based on polyether ketone (BioHPP) were also proposed for denture base fabrication [9].

Recently, new digital technologies have increasingly been used in dentistry and are also available for denture base fabrication, both subtractive and additive [3,6,12,13,14,15]. When compared with heat-cured PMMA, digital technologies can theoretically accelerate the fabrication of the denture, reduce the possibility of errors, improve precision and achieve better material properties. It is also possible to reduce the number of patient visits to the dental office and reduce the dental technician’s working time [1,16,17,18]. With stored computer data, it is easy to reproduce the same denture if necessary [19,20]. However, there is a lack of scientific data for these types of materials, especially for additive technologies.

The denture base should be meticulously fitted to the residual ridge after its fabrication. Due to the resorption of the alveolar bone, which is a chronic, progressive and irreversible process, the shape of the residual ridge changes. Denture relining, as a common clinical procedure in dentistry, can prolong the use of the existing denture by adapting the denture base to the changes in soft and hard tissues. This procedure is much faster and less expensive than fabricating a new denture. Hard and soft denture liners can be used. Soft denture liners can be used for both short- and long-term use, and they can be silicone- and acrylate-based [21,22].

Soft denture liners have a cushioning effect and can contribute to an even distribution of functional loads on the denture-bearing area and improve patient comfort, especially in cases of undercuts, sensitive mucosa, and bruxomania [22,23,24]. They may also be helpful after surgical procedures and for immediate dentures. It is proven that soft denture lining can improve oral-health-related quality of life, masticatory function, and overall patient satisfaction with the denture [25,26,27].

Bond strength between the denture base material and soft denture liner is considered as one of the key factors for the long-term success of the relining procedure [23,28,29,30,31,32,33]. However, the bond strength values between denture base materials (especially in additive and subtractive manufacturing) and soft denture liners are poorly studied, and standardised tests are rarely used. Therefore, the aim of this study was to investigate the tensile bond strength values between denture base materials and soft denture liners using the method described in specification ISO No. 10139-2:2016 [34], with an emphasis on denture base materials for computer-aided design–computer-aided manufacturing (CAD-CAM) technology. Additionally, the type of failure was investigated. The null hypothesis was stated: There is no difference in tensile bond strength between the different denture base materials and soft denture liners, and there is no difference in the type of failure between the different denture base materials and soft denture liners.

## 2. Materials and Methods

Seven different denture base materials and two different soft denture liners were used in this study. The materials used are shown in Table 1.

This study was performed according to the specification ISO No. 10139-2:2016 [34]. Plates with dimensions of 25 ± 3 mm × 25 ± 3 mm and a thickness of 3 ± 0.5 mm, composed of denture base material, were the basis for specimen preparation. The flat surfaces of the plates were kept plane-parallel and wet-ground with standard P500 metallographic grinding paper. After the preparation of the plates, they were stored in a water bath at 37 ± 1 °C for 30 ± 2 days.

Two plates, a polytetrafluoroethylene (PTFE) collar and a PMMA rod, were needed for one specimen. The PTFE collar had an inner diameter of 10 ± 0.5 mm and a height of 3 ± 0.25 mm. The PMMA rod had an outer diameter of 10 mm and a height of 20 mm.

After the plates were removed from the water bath, they were dried, and for the silicone-based liner, adhesive was applied to the adhesive surface of the plate. The PTFE collar was placed in the centre of the plate, and the prepared soft liner material was applied with slight excess while being confined within the PTFE collar and closed with the second plate. The specimen was clamped for 1 h. Then, the PMMA rod was attached to the top of the second plate using cyanoacrylate cement. A custom-made template was used to assemble the specimen and to maintain the vertical alignment of the specimen.

The specimens were again stored in a water bath at 37 ± 1 °C for 23 ± 1 h. Immediately after removal from the water bath, the specimens were placed in the universal testing machine (Autograph AGS-X, Shimadzu, Kyoto, Japan). To ensure the vertical alignment of the specimen in the testing machine, a custom-made loading assembly was used. The tensile test was performed at a displacement rate of 10 mm/min (Figure 1). The maximum load (F) during debonding was recorded. The sample size was determined using the specification ISO No. 10139-2:2016 [34]. For each denture base material in combination with one soft liner, 10 specimens were prepared, and 140 measurements were performed in total.

The tensile bond strength B (MPa) was calculated according to the formula B = F/A, where F (N) is the maximum load recorded and A (mm^2^) is the adhesive area. The adhesive area was defined according to the inner diameter of the PTFE collar.

The type of failure was determined visually according to the instructions of specification ISO 10365:2022 [35]. High-resolution photographs were taken using a digital single-lens reflex camera EOS 250D (Canon, Ota City, Tokio, Japan) with a macro-objective at a 10× magnification. A distinction was made between adhesive, cohesive and mixed types of failure.

MedCalc^®^ Statistical Software v20.010 was used for the statistical analysis. The Kruskal–Wallis test with Dunn’s post hoc test was used to analyse the values of tensile bond strength, and Fisher’s exact test was used to analyse the type of failure. *p* Values < 0.05 were considered statistically significant.

## 3. Results

The results of tensile bond strength for both soft denture liners are shown in Table 2 and in Figure 2 and Figure 3. Two graphs presenting tensile stress as a function of strain for both soft denture liners used in this study are shown in Figure 4 and Figure 5.

There was no statistically significant difference in the tensile bond strength values between the GC Soft Liner and different denture base materials (*p* ˃ 0.05) (Table 2).

Statistically significantly different values of tensile bond strength were found between the GC Reline II Soft and different denture base materials (*p* ˂ 0.05) (Table 2).

The tensile bond strength value between the heat-cured PMMA denture base material and GC Reline II Soft was statistically significantly higher (*p* ˂ 0.05) than the bond strength values between the GC Reline II Soft and both additive-manufactured materials (Table 2).

The bond strength value between the poliamide denture base material and GC Reline II Soft was statistically significantly lower (*p* ˂ 0.05) than the bond strength values measured between the GC Reline II Soft and all three materials used for subtractive denture fabrication and between the GC Reline II Soft and heat-cured PMMA material (Table 2).

The tensile bond strength values between all three subtractive-manufactured denture base materials and the GC Reline II Soft were statistically significantly higher (*p* ˂ 0.05) than the tensile bond strength values between the GC Reline II Soft and poliamide, as well as both additive-manufactured denture base materials (Table 2).

In addition, a statistically significantly higher tensile bond strength value was found between the Polident pink CAD-CAM and GC Reline II Soft compared to the tensile bond strength value between the Anaxdent pink blank and GC Reline II Soft (*p* ˂ 0.05) (Table 2).

Both additive-manufactured denture base materials showed statistically significantly lower tensile bond strength values (*p* ˂ 0.05) in combination with GC Reline II Soft compared to the combination of GC Reline II Soft with all three subtractive-manufactured denture base materials and with heat-cured PMMA (Table 2).

The results for the type of failure for both soft denture liners are shown in Table 3. Representative photographs of the fracture modes are shown in Figure 6.

For the GC Soft Liner, there was a statistically significant difference in the results for the type of failure between different denture base materials (*p* ˂ 0.05). When the GC Soft Liner was combined with polyamide and both additive denture base materials, the type of failure was predominantly adhesive, whereas when the GC Soft Liner was combined with all other denture base materials, the type of failure was predominantly or exclusively cohesive (Table 3).

For the GC Reline II Soft, there was also a statistically significant difference between the results for the type of failure for different denture base materials (*p* ˂ 0.05). For the combination of GC Reline II Soft and Ivobase CAD pink, the type of failure was dominantly adhesive, and for the combination of GC Reline II Soft and the Imprimo LC denture, the type of failure was predominantly cohesive (Table 3).

## 4. Discussion

Denture relining is a clinical procedure used to adjust the denture base to soft and hard tissue changes. It extends the use of the existing denture, is less expensive and faster than fabricating a new denture and improves the patient’s oral-health-related quality of life, masticatory function and overall satisfaction with their denture. The bond strength values between denture base materials manufactured with digital technologies (especially additive manufacturing) and denture liners are poorly studied.

Therefore, the aim of this study was to investigate tensile bond strength values between denture base materials and soft denture liners, with an emphasis on denture base materials for CAD-CAM technology. Three different materials for subtractive manufacturing and two for additive manufacturing were included in our study. In addition, a polyamide denture base material was included, because it is used as an alternative to PMMA in standard analogue processes for denture base fabrication. Heat-cured PMMA was included as the gold standard among the materials used for denture base fabrication.

According to the results, there was no statistically significant difference in the tensile bond strength values between the different denture base materials and soft liner, but there was a statistically significant difference in the tensile bond strength values between the different denture base materials and Reline II Soft. The polyamide and both additive-manufactured denture base materials showed statistically lower tensile bond strength values when combined with the GC Reline II Soft compared to the heat-cured PMMA and all three subtractive-manufactured denture base materials. There was also a statistically significant difference in the tensile bond strength values between the Polident pink CAD/CAM and Anaxdent pink blank in combination with the Reline II Soft. Therefore, for the results of tensile bond strength between the GC Soft Liner and different denture base materials, the null hypothesis was accepted. For the tensile bond strength results between the Reline II Soft and different denture base materials, the null hypothesis was rejected.

In terms of the type of failure, both the additive-manufactured and polyamide denture base materials showed statistically significantly different values for the GC Soft Liner, with the adhesive type dominating. Therefore, the null hypothesis for the results regarding the type of failure between the GC Soft Liner and different denture base materials was rejected. The GC Soft Liner is acrylate-based and used without adhesive, because it is considered that the monomer of the liner causes the swelling of the surface of the denture base material and the chemical bond between the two materials. Polyamide materials have a different chemical composition from PMMA, also being additive-manufactured materials that are not pure PMMA materials in terms of composition, and they have many other additives. Therefore, it can be concluded that there was no chemical bonding, which was the cause of the predominantly adhesive type of failure [33].

In our study, a statistically significant difference in the type of failure was found between the GC Reline II Soft and the different denture base materials. For the combination of GC Reline II Soft and Ivobase CAD pink, the type of failure was predominantly adhesive, and for the combination of GC Reline II Soft and the Imprimo LC denture, the type of failure was predominantly cohesive. No dominant or exclusive type of failure was observed for the combination of GC Reline II Soft and all other denture base materials. The null hypothesis for the failure type between GC Reline II Soft and the different denture base materials was rejected.

In the study conducted by Awad et al. [36], the tensile bond strength between denture base materials and denture liners was investigated. The tensile bond strength values varied between different material combinations, and the type of failure between the denture base materials and soft liners was predominantly adhesive. Wemken et al. [37] found no statistically significant difference in the tensile bond strength values between different denture base materials (heat-cured PMMA, subtractive and additive manufacturing) and a soft liner, while the type of failure was exclusively adhesive. Azpiazu-Flores et al. [38] described the lowest tensile bond strength values between additive-manufactured denture base materials and long-term soft liners. In contrast, Choi et al. [31] described the lowest tensile bond strength values between subtractive-manufactured denture base materials and a soft liner, with most cases showing the adhesive type of failure.

Our results for the type of failure are partially in accordance with the results obtained by Awad et al. [36]. In their study, they included two soft denture liners, both acrylate-based. For one, the results were similar, while for the other, the results differed. Wemken et al. [37] included just one soft denture liner, which was silicone-based, in their study and observed an exclusively adhesive type of failure for all the denture base materials, in contrast with our results. Choi et al. [31] included one acrylate-based and two silicone-based soft denture liners in their study and observed an exclusively or predominantly adhesive type of failure in all combinations of the materials, again in contrast with our results. When comparing our study with the aforementioned studies, it can be noted that the materials used were from the same groups of materials but not from the same manufacturers. Additionally, the preparation of the samples and testing methods differed greatly, which could be the reason for such a discrepancy in the results for the type of failure. No firm conclusions can be drawn at this point, and further investigations are required considering the type of failure.

A polyamide denture base material was included in this study because it is used as an alternative to PMMA in the standard fabrication of analogue denture bases. It has a crystalline structure, making it a more chemical-resistant material that does not react with adhesives and monomers, in contrast to PMMA [39,40]. Therefore, it is more difficult to achieve a satisfactory bond strength with soft denture liners, and it is recommended that one uses additional surface preparation methods for polyamide materials [41,42].

In ISO 10139-2:2016 [34], it is stated that it is important to achieve a vertical alignment of the specimen in the testing machine to avoid torsional forces acting on the specimen. For this reason, we used a custom-made loading assembly with a flexible connection in the upper part of the assembly. Another way to achieve vertical alignment was demonstrated by Kim et al. [21], using a ball-and-socket joint in the lower part of the assembly.

ISO 10139-2:2016 [34] also states that the minimum bond strength required for soft long-term denture liners should be at least 1.0 MPa for soft materials and at least 0.5 MPa for extra-soft materials for at least 8 of the 10 specimens tested. The GC Reline II Soft, which we used in our study, met the minimum requirements for all denture base materials. The GC Soft Liner, on the other hand, had results all below 0.3 MPa, but according to the manufacturer, it is a short-term soft liner; thus, it does not need to meet the minimum requirement of 1.0 MPa. For short-term soft liners, ISO 10139-2:2016 does not specify minimum bond strength requirements. It should also be mentioned that some authors have cited 0.44 MPa as a minimum requirement for soft liners in previous studies [23,31,43,44].

It is stated in the literature that acrylate-based soft liners have a higher bond strength than silicone-based materials. This is due to the similar chemical compositions of the denture base material and the soft liner, which allow for a chemical bond between the two materials and better adhesion. When using silicone-based soft liners, it is important to use a suitable adhesive; otherwise, no chemical bond between the two materials will be established [28]. In our study, the silicone-based soft liner showed statistically significantly higher values for tensile bond strength than the acrylate-based one, but since the acrylate-based material in our study is intended for short-term use and the silicone-based one is intended for long-term use, they cannot be directly compared.

It was observed that during the mixing of the GC Soft Liner, there were many air inclusions inside the material, and these inclusions were also observed on the contact surface of the denture base material and soft liner. This material is mixed by hand, and these air inclusions reduce the contact area between the two materials and decrease the bond strength. The GC Reline II Soft is mixed using mixing tips, so that there are no visible air inclusions inside the mixed material, and with careful application, air inclusion on the contact surface of the two materials can be avoided. Kim et al. [21] pointed out this problem in their study.

In previous studies [31,45], micropores and air inclusions were found on the contact surface of a denture base material and silicone-based soft liner using a scanning electron microscope (SEM), which were not visible to the naked eye. Since the primer is used to ensure the adhesion of the silicone-based material, it is assumed that either the chemical reaction between the solvent in the primer (ethyl acetate) and the denture base material or the evaporation of the solvent is the cause of these air inclusions, and they may act as the fracture initiation site and reduce the contact area and the bond strength.

According to data from the available literature, it can be observed that in previous studies, different investigation methods were used for bond strength tests, including shear bond, peel bond and tensile bond strength tests, while the tensile bond strength test was most commonly used [37]. Tensile strength testing was also performed in different ways, with different specimen preparation methods, different specimen surface preparation techniques and different displacement rates. It can be concluded that the main problem in tensile bond strength testing was the control of the adhesive surface. Therefore, different specimen preparation methods were used. Some authors used the method with a metal flask and specimen invested in putty silicone impression material. The specimens were usually rod-shaped, with free space for the soft liner between two parts. The vertical orientation of the specimen was also controlled in this way. After the soft liner hardened, the specimens could be easily removed from the dental flask. Other authors used the method described in ISO 10139-2:2016 [34], but it was usually modified. In this method, the bonding surface is controlled with a PTFE or PE collar between the two plates of the denture base material. Since different examination methods were used in previous studies, it is difficult or not possible to compare different studies. It is only possible to draw certain conclusions within a single investigation. Therefore, in our investigation, we aimed to follow the instructions of the specification ISO 10139-2:2016 in full [34].

A general statement about the bond strength between additive and subtractive denture base materials, on the one hand, and soft denture liners, on the other, is currently not possible for several reasons. First, there are only a few studies that have been conducted on this topic. Second, different test methods were used in these studies. Third, different materials were used as a control group (heat-cured, injection-moulded PMMA from different manufacturers). Fourth, the research results vary between different studies; thus, the results cannot be summarised, and no clear conclusions can be drawn.

The limitation of this study is the fact that only one acrylate-based and one silicone-based soft denture liner were used, and for more firm conclusions to be obtained, more soft lining materials should be included in future investigations.

Our proposition for future investigations is to include more different soft denture lining materials from all categories, including those for short-term and long-term use, both acrylate-based and silicone-based, so that more firm conclusions could be obtained. Additionally, it should be investigated whether different types of surface pretreatments for additive-manufactured dentures could improve the tensile bond strength values when soft denture liners are used.

## 5. Conclusions

The use of digital technologies in denture base fabrication may influence the tensile bond strength values between denture base materials and soft denture liners (with different types of failure) when compared with heat-cured PMMA denture base materials. 

There is no significant difference in tensile bond strength between the acrylate-based soft denture liner and denture base materials, which is not the case for the silicone-based soft denture liner. For the silicone-based soft denture liner used in combination with both additive-manufactured denture base materials, the values of tensile bond strength were statistically significantly lower than those for the same material used in combination with heat-cured PMMA and all three subtractive-manufactured denture base materials. The basic Polident pink CAD-CAM disc showed the highest tensile bond strength value in combination with the silicone-based soft liner.

Based on the higher values of tensile bond strength between subtractive-manufactured denture bases and PMMA denture bases with silicone-based soft liners, it can be suggested that practitioners use this combination of materials more frequently. All the investigated denture base materials can be combined well with acrylate soft denture liners.

## Figures and Tables

**Figure 1 materials-16-04615-f001:**
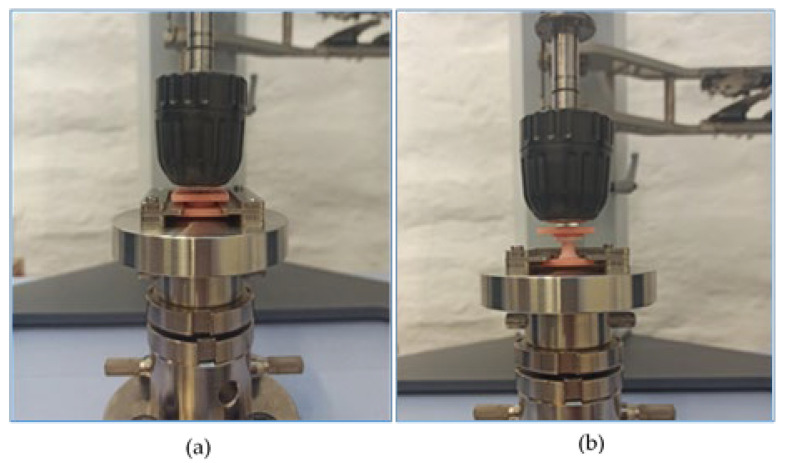
Specimen placed in the universal testing machine at the beginning of testing (**a**) and during testing (**b**).

**Figure 2 materials-16-04615-f002:**
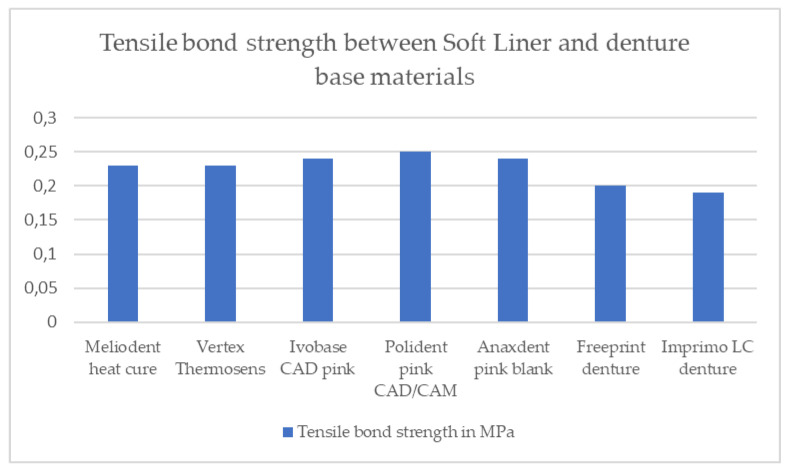
Tensile bond strength results between soft liner and different denture base materials.

**Figure 3 materials-16-04615-f003:**
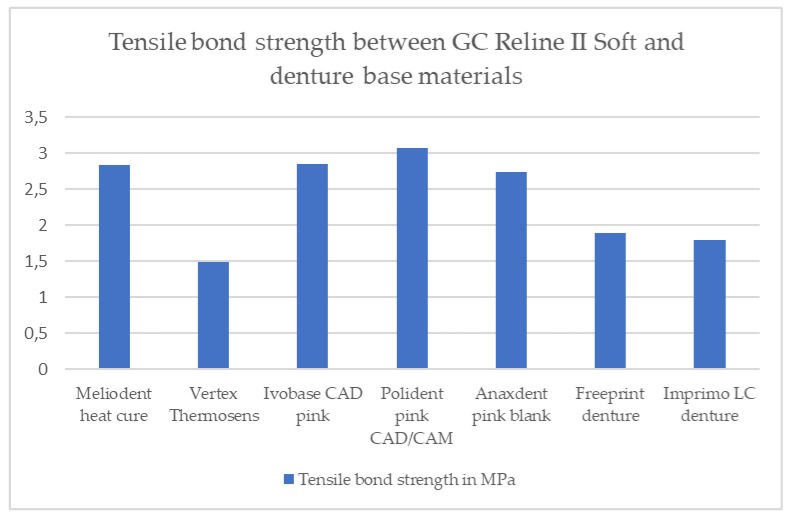
Tensile bond strength results between GC Reline II Soft and different denture base materials.

**Figure 4 materials-16-04615-f004:**
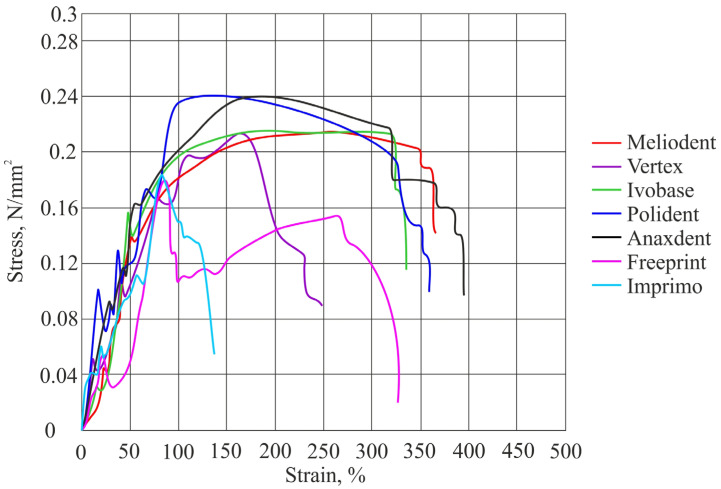
Graph showing tensile stress as a function of strain with average values obtained for the soft liner.

**Figure 5 materials-16-04615-f005:**
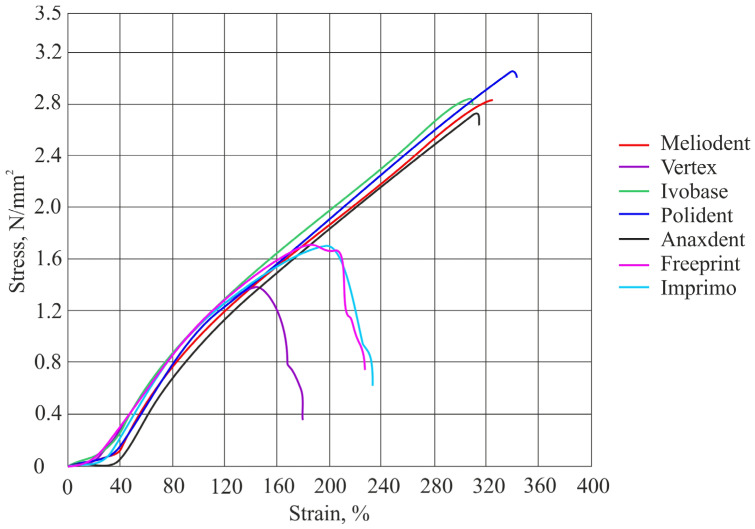
Graph showing tensile stress as a function of strain with average values obtained for the for Reline II Soft.

**Figure 6 materials-16-04615-f006:**
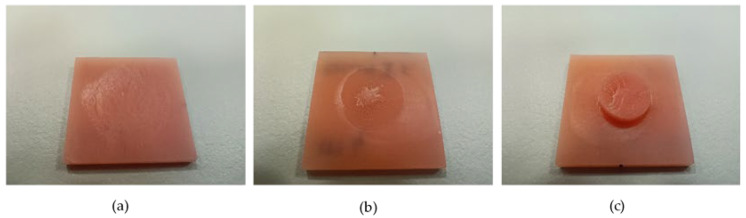
Representative photographs of failure types: adhesive type of failure (**a**), mixed type of failure (**b**) and cohesive type of failure (**c**).

**Table 1 materials-16-04615-t001:** Materials used in this study.

Name of the Material	Manufacturer	Description and Purpose of the Material
Meliodent heat cure	Kulzer, Hanau, Germany	Denture base material, PMMA, heat-cured
Vertex Thermosens	Vetex Dental, Soesterberg, Netherlands	Denture base material, polyamide, injection technique
Ivobase CAD pink V	Ivoclar Vivadent, Schaan, Liechtenstein	CAD CAM denture base material, subtractive manufacturing
Polident pink CAD/CAM disc basic	Polident d.o.o., Volčja draga, Slovenia	CAD CAM denture base material, subtractive manufacturing
Anaxdent pink blank U medium pink	Anaxdent GmbH, Stuttgart, Germany	CAD CAM denture base material, subtractive manufacturing
Freeprint denture	Detax, Ettlingen, Germany	CAD CAM denture base material, additive manufacturing
Imprimo LC denture	Scheu, Iserlohn, Germany	CAD CAM denture base material, additive manufacturing
Soft liner	GC Europe, Leuven, Belgium	Soft denture liner, acrylate-based, direct relining method
Reline II soft	GC Europe, Leuven, Belgium	Soft denture liner, silicone-based, direct relining method

**Table 2 materials-16-04615-t002:** Tensile bond strength between denture base materials and soft liners.

	GC Soft Liner		GC Reline II Soft	
	Mean (MPa)	SD	Mean (MPa)	SD
1. meliodent heat cure	0.23	0.07	2.84 ^2, 6, 7^	0.33
2. vertex thermosens	0.23	0.06	1.49 ^1, 3, 4, 5^	0.47
3. ivobase cad pink	0.24	0.08	2.85 ^2, 6, 7^	0.23
4. polident pink cad/cam	0.25	0.08	3.07 ^2, 5, 6, 7^	0.23
5. anaxdent pink blank	0.24	0.08	2.74 ^2, 4, 6, 7^	0.27
6. freeprint denture	0.20	0.07	1.89 ^1, 3, 4, 5^	0.47
7. imprimo lc denture	0.19	0.09	1.80 ^1, 3, 4, 5^	0.50

MPa = megapascal, SD = standard deviation. Superscripted numbers indicate a statistical difference between the groups of denture base materials, *p* < 0.05.

**Table 3 materials-16-04615-t003:** Type of failure.

	Soft Liner			GC Reline II Soft		
	Type of Failure			Type of Failure		
	Adhesive	Cohesive	Mixed	Adhesive	Cohesive	Mixed
meliodent	1	9	0	4	6	0
vertex thermosens	7 *	3 *	0 *	6	1	3
ivobase cad pink	0	9	1	8 *	2 *	0 *
polident pink cad/cam	0	10	0	3	6	1
anaxdent pink blank	0	10	0	6	4	0
freeprint denture	7 *	3 *	1 *	0 *	5 *	5 *
imprimo lc denture	9 *	1 *	0 *	0	8	2

* indicates statistical difference between groups (*p* < 0.05).

## Data Availability

Not applicable.

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
