# Peer review of "Tensile Bond Strength between Different Denture Base Materials and Soft Denture Liners"

_materials, 2023, doi:10.3390/ma16134615_

Round 1

Reviewer 1 Report

Dear authors,

please find below my suggestions

Best Regards

Minor editing of English language required

Author Response

Dear sir or madam,

We wish to thank you for taking the time to review and evaluate our manuscript. We addressed all the concerns that were raised and we made the revisions in the manuscript according to the provided suggestions. Here is a point-by-point response to the comments and concerns.

Comment 1: The introduction needs to be reinforced. The authors should compare the materials object of study, with other materials in the literature used as dental prostheses.

Response: We tried in the introduction to emphasize the advantages and disadvantages of the materials used in the study, and as suggested we mentioned other materials and processing techniques for denture base fabrication.

Comment 2: The title concerns the tensile bond strength between different denture base materials and soft denture liners, but no graph is given. The figures alone showing the test machine used are not sufficient.

Response: In the results section we added Figure 2 and Figure 3 with the graphs showing the results.

Comment 3: Please process the data obtained and add related graphs.

Response: We added Figure 4. with the graphs showing tensile stress as a function of strain with average values for each denture base material and separately for two soft denture liners.

Comment 4: The conclusions are too general. Add future perspectives.

Response: We revised the conclusion section to obtain more distinct conclusions, and at the end of the discussion section we added a proposition for future investigations.

We look forward to hear from you regarding our submission and to respond to any further questions and comments.

With regards

The authors

Reviewer 2 Report

The authors have evaluated the tensile bond strength between CAD-CAM denture base materials and denture soft liners. Few suggestions are:

1. The result section of the abstract must report the numerical values achieved. 

2. The introduction part is concise; can further be improved using the following relevant papers:

Khan AA, Fareed MA, Alshehri AH, Aldegheishem A, Alharthi R, Saadaldin SA, Zafar MS. Mechanical properties of the modified denture base materials and polymerization methods: a systematic review. International Journal of Molecular Sciences. 2022 May 20;23(10):5737.

Khan AA, De Vera MA, Mohamed BA, Javed R, Al‐Kheraif AA. Enhancing the physical properties of acrylic resilient denture liner using graphene oxide nanosheets. Journal of Vinyl and Additive Technology. 2022 Aug;28(3):487-93.

3. From lines 78-91, its a too long sentence. Please rephrase the text.

4. In Table 2, failure spelling is wrong

5. Please suggest some future directions at the end of the Discussion part for continuity of this research

6. The conclusion part is just a repetition of the Results. What is the take-home lesson? Please rewrite this part

Author Response

Dear sir or madam,

We wish to thank you for taking the time to review and evaluate our manuscript. We addressed all the concerns that were raised and we made the revisions in the manuscript according to the provided suggestions. Here is a point-by-point response to the comments and concerns.

Comment 1: The result section of the abstract must report the numerical values achieved.

Response: We added numerical values obtained in the result section of the abstract.

Comment 2: The introduction part is concise; can further be improved using the following relevant papers..

Response: we revised the introduction section and supplemented it by mention other denture base materials and processing techniques.

Comment 3: From lines 78-91, its a too long sentence. Please rephrase the text.

Response: we removed the mentioned sentence and we replaced it with Table 1 with the list of all materials used in the study, as was suggested by other reviewer.

Comment 4: In Table 2, failure spelling is wrong.

Response: we corrected the spelling in the Table 2 for the word failure, and corrected the same mistake with the spelling of the same word through the whole manuscript.

Comment 5: Please suggest some future directions at the end of the Discussion part for continuity of this research.

Response: we added the proposition for future investigations at the end of discussion section.

Comment 6: The conclusion part is just a repetition of the Results. What is the take-home lesson? Please rewrite this part.

Response: we rewrote the conclusions section and we added a direct proposition for the practitioners.

We look forward to hear from you regarding our submission and to respond to any further questions and comments.

With regards

The authors

Reviewer 3 Report

Title: good   Abstract: - Please add the statistical test - L25: how can influence? positive? negative, more details - Please correct failure among the text   Introduction: - L69: poorly studied: it is not sufficient for the originality, please clarify the originality of the paper   Methods: - L78-90: please make a table to clarify - L93: +/- 3 mm it is huge, or please add a reference - How many samples were used for each group? - P500, Why? any reference? - Please correct alignment among the text - 10 mm/min? it is so fast, any reference? - 10 measurement? 10 samples for each? or the same sample with 10 measurements? - Fig 1 and 2 could be merged - mm2 please correct - Type of failure, please add the classification with a reference and the complete name and company of the microscope with the magnifications - Sample size test should be performed   Results: - Some images for the different failure types should be added   Discussion: - please discuss the failure types - SEM? please sue the complete name not only the abbreviations - Please clarify the limitations of the present study   References: - Please follow the MDPI style

Author Response

Dear sir or madam,

We wish to thank you for taking the time to review and evaluate our manuscript. We addressed all the concerns that were raised and we made the revisions in the manuscript according to the provided suggestions. Here is a point-by-point response to the comments and concerns.

Comment 1: Abstract: - Please add the statistical test

Response: we added the statistical test used in the study at the end of materials and methods in the abstract.

Comment 2: L25: how can influence? positive? negative, more details

Response: we revised the conclusions in the abstract section.

Comment 3: Please correct failure among the text.

Response: we corrected the mistake with the spelling for word failure through the whole manuscript.

Comment 4: Introduction: - L69: poorly studied: it is not sufficient for the originality, please clarify the originality of the paper.

Response: we revised the introduction section, and we tried to stress few things: as it is known to us there are just a few studies that investigated the tensile bond strength between soft liners and additive and subtractive manufactured denture base materials, those studies we qouted in the discussion section. As it is known to us, this is the first study using the instructions from ISO specification 10139-2 with additive manufactured denture base material included in the investigation.

Comment 5: Methods: - L78-90: please make a table to clarify.

Response: as suggested, we added Table 1 with the list of the all materials used in the study and removed the sentence in the L78-91.

Comment 6: L93: +/- 3 mm it is huge, or please add a reference

Response: the size and the tolerance of the size of the plate are the exact directions from ISO 10139-2.

Comment 7: How many samples were used for each group?

Response: 10 samples were used in each group, and we revised the materials and methods section to clarify that.

Comment 8: P500, Why? any reference?

Response: the use of P500 metallographic grinding paper is the requirement from ISO 10139-2.

Comment 9: Please correct alignment among the text

Response: we corrected the alignment of the text, tables and figures through the manuscript.

Comment 10: 10 mm/min? it is so fast, any reference?

Response: the displacement rate is determined by the instructions from ISO 10139-2.

Comment 11: 10 measurement? 10 samples for each? or the same sample with 10 measurements?

Response: we revised the materials and methods section to clarify that. We prepared 10 samples for each group with 1 measurement for 1 sample, 140 measurements in total.

Comment 12: Fig 1 and 2 could be merged

Response: we made the change as suggested.

Comment 13: mm2 please correct

Response: we corrected the mistake with superscript.

Comment 14: Type of failure, please add the classification with a reference and the complete name and company of the microscope with the magnifications

Response: the sentence in materials and methods section:” Type of failure was determined using optical microscopy”, was a mistake ocurred writing the article that we unintentionaly overlooked, and we regret it. Type of failure was determined visually according to instructions from ISO ISO 10365:2022. High-resolution photographs were taken using a digital single-lens reflex camera EOS 250D (Canon, Ota City, Tokio, Japan) with a macro objective at a 10x magnification. We revised materials and methods section accordingly.

Comment 15: Sample size test should be performed

Response: the sample size was determined by the instructions for the sample size from the ISO 10139-2.

Comment 16: Results: - Some images for the different failure types should be added 

Response: we added Figure 5. with representative photographs of failure types.

Comment 17: Discussion: - please discuss the failure types

Response: we expanded the discussion section by comparing our results for the type of failure with previous studies.

Comment 18: SEM? please sue the complete name not only the abbreviations

Response: we corrected the mistake in the manustcript as suggested.

Comment 19: Please clarify the limitations of the present study

Response: we revised the discussion section and clarified the limitations of the present study.

Comment 20: References: - Please follow the MDPI style

Response: we checked and revised reference list to follow the MDPI style.

We look forward to hear from you regarding our submission and to respond to any further questions and comments.

With regards

The authors

Round 2

Reviewer 2 Report

No more comments

minor English editing is required

Reviewer 3 Report

Good answers and revision